# Analysis of Biochemical and Genetic Variability of *Pleurotus ostreatus* Based on the β-Glucans and CDDP Markers

**DOI:** 10.3390/jof8060563

**Published:** 2022-05-25

**Authors:** Marcel Golian, Zuzana Chlebová, Jana Žiarovská, Lenka Benzová, Lucia Urbanová, Lucia Hovaňáková, Peter Chlebo, Dana Urminská

**Affiliations:** 1Horticulture and Landscape Engineering Faculty, Institute of Horticulture, Slovak University of Agriculture in Nitra, Tr. A. Hlinku 2, 949 76 Nitra, Slovakia; xbenzova@uniag.sk; 2AgroBioTech Reseach Centre, Slovak University of Agriculture in Nitra, Tr. A. Hlinku 2, 949 76 Nitra, Slovakia; zuzana.chlebova@uniag.sk (Z.C.); xzamieskova@uniag.sk (L.U.); 3Faculty of Agrobiology and Food Resource, Institute of Plant and Environmental Sciences, Slovak University of Agriculture in Nitra, Tr. A. Hlinku 2, 949 76 Nitra, Slovakia; jana.ziarovska@uniag.sk (J.Ž.); xulicna@uniag.sk (L.H.); 4Faculty of Agrobiology and Food Resources, Institute of Nutrition and Genomics, Nitra, Slovak University of Agriculture in Nitra, Tr. A. Hlinku 2, 949 76 Nitra, Slovakia; peter.chlebo@uniag.sk; 5Faculty of Biotechnology and Food Sciences, Institute of Biotechnology, Slovak University of Agriculture in Nitra, Tr. A. Hlinku 2, 949 76 Nitra, Slovakia; dana.urminska@uniag.sk

**Keywords:** oyster mushroom, β-glucan, glucans, *Pleurotus ostreatus*, CDDP markers

## Abstract

Oyster mushroom (*Pleurotus ostreatus*) is still one of the most cultivated edible and medicinal mushrooms. Despite its frequent cultivation around the world, there is currently just a little information available on the variability of strains in terms of the content of β-glucans in them. This work presents an extensive study of 60 strains in terms of the content of α-glucans and β-glucans in their caps and stipes. The authenticity of the production strains based on an analysis of the variability of their genome by CDDP (Conserved DNA-derived polymorphism) markers was confirmed, whereas identical CDDP profiles were identified between samples 45, 89, 95, and 96. Genetic variability of the analyzed production strains showed a high polymorphism and effective discriminative power of the used marking technique. Medium positive correlations were found among the CDDP profiles and β-glucan content in the group of strains that generated the same CDDP profiles, and low negative correlation was found among these profiles in the group of low β-glucan content strains. For the determination of glucans content, Mushroom and Yeast analytical enzymatic kit (Megazyme, Bray, Co. Wicklow, Ireland) were used. The results clearly showed that the stipe contains on average 33% more β-glucans than the cap. The minimum detected β-glucan content in the stipe was in strain no. 72, specifically 22%, and the maximum in strain no. 43, specifically 56%, which after the conversion represents a difference of 155%. From the point of view of β-glucan content, the stated strain no. 43 appears to be very suitable for the commercial production of β-glucans under certain conditions.

## 1. Introduction

β-glucans are heterogeneous polysaccharides extracted from organic sources such as cereals, mushrooms, seaweeds, and microorganisms [1,2,3]. Glucans have a number of unique characteristics, some of them are described below. The greatest importance of glucans is their prebiotic function [4,5]. Due to the structure, there are two primary underlying kinds of polysaccharides, α-glucans (e.g., glycogen, starch) and β-glucans, which do not contain starch [3,4,6,7,8]. α-1,3-glucans are semi-crystalline, water-insoluble polysaccharides tracked down normally in mushrooms [9]. β-glucans are made up of the main β-1,3-glucan backbone. The main chain, depending on the source, is branched by β-1,4 or β-1,6-glucans, or is not branched at all. Chains are linked through β-glycosidic bonds [1,3]. They also form substrates for enzymes such as endo-β-1,3-glucanase [10]. In every fungal organism there is a different amount of β-glucans depending on the type of fungus [11]. The important role of biological activities in fungi—among others of equal importance (e.g., polyphenols)—is performed by β-glucans. They are the main components of fungal cell walls of mushrooms, participate in the protection of the cell from external influences, and are involved in structural and energetic storage issues [12,13]. It has been proven by experiments on animals that there is no possibility for β-glucans overdose [1]. In countries such as USA, Canada, Sweden, Finland, and China, β-glucans have a license as pharmaceutical reagents [14]. The human body can absorb 0.5–4.9% of soluble β-glucans. The quality of β-glucans is influenced by the technological process of processing a specific food product that contains them [1,3]. β-glucans are synthetized in cell walls. The synthesis of β-glucans can be inhibited by the application of steroidal alkaloid products. It results in the digestion of the fungal cell wall and biosynthesis is stopped [15]. Destroying β-glucan bonds is also possible by hydrolysis using endo-β-1,3-glucanase [10]. These findings are highly applicable in the development of antifungal agents, in research of genetics, and physiology of fungi [12]. It has been demonstrated that β-glucans are capable of self-assembling into a variety of nanocomposite biomaterials. Nanocomposite biomaterials are then used in obtaining the diagnosis and subsequent therapy in human [5,16,17]. Up to now, scientific researches have confirmed the beneficial effect of glucans on the human body and even the ability of glucans to suppress or totally cure the disease. According to Sadoughi et al. [18], the potential of glucans is that they have an anti-inflammatory effect and can cause cell death, known as apoptosis, and therefore, they have an anticancer effect. Based on another study, β-glucans control the levels of glucose and insulin in blood, decrease the levels of cholesterol, and thus reduce the risks of various lifestyle-related diseases [1,19]. Dextran turns out to be significant in the manufacture of medical devices (wound dressings) because it helps in the treatment of wounds [20]. β-glucans are well known as a food supplement, which supports immune system and antioxidant activity, and improves digestion thanks to prebiotic activity. The immune response happens when β-glucan attaches to specific receptors of an immune cell and induces phagocytosis, a biological mechanism that keeps tissue equilibrium [21]. This effect happens directly. The second effect of β-glucans, which is indirect, is happening in the human digestive tract, more precisely in the colon. β-glucans as prebiotics are used by colon microflora for producing SCFAs (short-chain fatty acids) [22]. Another significant effect is that they prevent fat absorption from food and stabilizing blood sugar (glucose) levels after consumption [3,23]. Due to the immunostimulatory properties of β-glucans, the planning of vaccine adjuvants based on β-glucans is an interesting perspective [24]. Mushrooms are an interesting component of human nutrition due to all of these properties and their high content of essential nutrients such as proteins, polysaccharides, and lipids [25]. The β-glucan, which is found in *Pleurotus ostreatus*, is called Pleuran. It was found that it enhances the weakened immunity in people with breast cancer [26].

Regardless of the financial significance of *Pleurotus ostreatus*, only a few genetic studies have been conducted due to methodological problems such as directed crosses studies between strains and, in many instances, incomplete or antagonistic data about the genetic material, its size, and organization features. Genetic linkage maps were constructed for this species based on the RAPD and RFPL markers and based on the repetitive DNA [27]. Classical DNA markers such as RAPD were applied in studies that describe the genetic variability of oyster mushrooms by different authors [28,29] or ISSR markers that were used in the study of Lin et al. [30]. New approaches, such as sequence analysis of ITS, translation elongation factor, and the second largest subunit of RNA polymerase II were used to analyze the genetic diversity of *Pleurotus ostreatus* strains by Liu et al. [31]. CDDP marker technique [32] is based on the presence of conserved DNA regions within plant/fungi genes involved in response to stresses and plant development, which are abundant as short conserved gene sequences that are present at multiple sites within the genomes. This method is universal and was applied in many species to generate polymorphic characteristics [33,34,35,36].

The presented work is focused on the determination of glucans content and identification of the differences in the glucans content of the stipes and caps, while the analysis was performed in 59 strains of *Pleurotus ostreatus*. The authenticity of the analyzed strains was verified by the CDDP analysis.

## 2. Materials and Methods

### 2.1. Biological Material

The biological material was acquired through international cooperation between universities and research institutes (Slovak University of Agriculture in Nitra, Slovakia; Crop Research Institute Czech Republic; Czech University of Life Sciences Prague, Czech Republic). Most of the cultures were provided from the culture collection of microorganisms from the Crop Research Institute, The National Program on Conservation and Utilization of Plant, Animal and Microbial Genetic Resources Important for Food and Agriculture (NPGR), supported by Ministry of Agriculture of the Czech Republic. The additional pure cultures of oyster mushroom (*Pleurotus ostreatus*) production strains were isolated from commercially available fruiting bodies from the local retail chains. The individual collection items of pure strain cultures were stored on PDA (potato dextrose agar) until used, and then inoculum was prepared from them. The list of production strains is given below (Table 1).

### 2.2. Growing Substrate

The growing substrate was created from a mixture of water and commercially available dry straw pellets pressed with high pressure from wheat straw.

For genetic analysis, young fruiting bodies of *Pleurotus ostreatus* were grown on a substrate that was created by a mixture of 2.6 parts of water and 1 part of straw pellets. So, the prepared mixture was filled into PE (polyethylene) food grade bags with a volume of 200 mL. This was followed by incubation at a temperature of 25 °C for 48 h. The heat treatment of the substrate was performed at a temperature of 60 °C in a climatic chamber KK 750 (POL-EKO-APARATURA sp. j., Wodzisław Śląski, Poland).

The substrates for the cultivation of fruiting bodies intended for the analysis of β-glucan content were prepared in a similar manner and on the same substrate as mentioned above. The dry straw pellets were mixed with water in an optimal ratio (dry pellets:water, in ratio 1:2.6). Subsequently, they were filled into 3500 mL transparent polyethylene buckets intended for food use. A suppression of possible microbial contamination and ensuring the selectivity of the growing substrate was performed by the incubation of the substrate at 25 °C for 48 h and subsequent heat treatment at 60 °C in climatic chamber KK 750 (POL-EKO-APARATURA sp. j., Wodzisław Śląski, Poland) for 24 h.

### 2.3. Inoculation, Incubation, Growth, and Processing of Fruiting Bodies

The inoculation of the substrates with the selected production strains took place by grained seedlings created from wheat overgrown with a pure culture of fungi. The amount of seedling was 5% by weight of the substrate. The inoculation itself was realized after heat treatment of the substrates, when they were cooled to 25 °C. Incubation of the inoculated substrates was realized at 25 °C and lasted until the complete overgrowth of the substrate by mycelium. The cultivation of fruiting bodies took place at 16 °C and optimum humidity (85% relative humidity).

For the analysis of the genetic polymorphism of the fruiting bodies, young fruiting bodies of the production strains, approximately 10 mm in size, were harvested (Figure 1). The fruiting bodies were frozen at −20 °C in Freezer BI540 ADV (Gorenje, Ljubljana, Slovenia) until further analysis.

The fruiting bodies for β-glucan analyses were harvested optimally in the optimal harvesting phase by harvesting in two growing waves (Figure 2). Each production strain was cultivated in four replicates and two cultivation cycles (spring and autumn production of year 2021). After harvesting, the fruiting bodies were cut and divided into stipes and caps. The stipe was separated from the hat at the point where the lamels began. The fruiting bodies were subsequently labeled and dried at 45 °C in a laboratory hot-air dryer Memmert UF 110 Plus (Memmert, Schwabach, Germany). Dry fruiting bodies were milled in a shear mill Retsch SM 100 (Retsch, Haan, Germany). So prepared material was stored in a dark and dry place.

### 2.4. Polymorphism Analysis of Pleurotus ostreatus by CDDP Markers

A total of 57 accessions of *Pleurotus ostreatus* were used for DNA-based marker analysis. Fresh material was transported to the laboratory and stored at −20 °C until further processing. A total genomic DNA was extracted according to Rogers and Bendich [38] protocol without any modifications. The quantity of extracted DNA was checked by Nanodrop (Implen GmbH, München, Germany) and the quality/functionality of extracted DNA was tested in PCR by using the universal ITS primers ITS1 plus ITS4 (data not shown) [39]. Different dilutions of extracted DNA were prepared for specific marker PCRs based on the results of ITS analysis that ranged from 1/9 up to 1/99.

Conserved DNA-derived polymorphism (CDDP) method was used to analyze the variability of *Pleurotus ostreatus* accessions. The single primer combination F plus R2b was used as reported by Collard and Mackill [32]. This primer combination was selected based on the previous tests of other reported primer combinations that produced only limited polymorphisms (data not shown). PCRs were prepared by EliZyme Robust HS (Elisabeth Pharmacon Ltd., Croydon Surrey, UK) together with the 400 nM of the each primer and the 50 ng of DNA in each reaction. Time and temperature profiling of the reaction was as follows: 95 °C for 5 min; 40 cycles of 95 °C for 45 s, 54 °C for 45 s, and 72 °C for 90 s; and 72 °C for 10 min. 

The obtained amplicons were separated in 1.5% agarose electrophoresis and the profiles of generated CDDP amplicons were converted into the 0–1 binary matrices using the GelAnalyzer software. Popgene software was used to calculate the characteristics of amplified CDDP loci (https://sites.ualberta.ca/~fyeh/popgene.html, accessed on 10 February 2022). The polymorphic information content (PIC) was calculated according to Smith et al. (1997) [40] using the formula: PIC = 1 − ∑ Pi2, where Pi is the frequency of the ith allele for all genotypes obtained. The UPGMA analysis was performed based on the Dice coefficient of genetic similarity (1945). The dendrogram was constructed in the DendroUPGMA free software (http://genomes.urv.cat/UPGMA/, accessed on 10 February 2022).

### 2.5. Determination of Glucans

As part of the β-glucan content evaluation, a total of 59 cap samples and 60 stipe samples of different strains of oyster mushroom (*Pleurotus ostreatus*) were experimentally evaluated. As part of the analysis, the average sample of all harvested fruiting bodies (approximately 45 adult fruiting bodies) for each specific production strain of *Pleurotus ostreatus* was evaluated.

For the determination of β-glucan content in the samples, β-glucan Assay Kit (Mushroom and Yeast) and K-YBGL analytical enzymatic kit from the company Megazyme (Megazyme, Bray, Co. Wicklow, Ireland) based on enzymatic hydrolysis and activity of oxidoreductases, namely exo-1,3-β-glucanase, β-glucosidase, glucose oxidase and peroxidase with the formation of the quinoneimine (colour substances) form of glucose oxidation, were used. The procedure for the determination of β-glucan in the samples was performed according to the methodical instructions of the manufacturer, which was given in the enclosed Mushroom and Yeast Beta-glucan Assay Procedure K-YBGL 11/19 with our minor methodological optimization. Within methodological optimization, 18 M H_2_SO_4_ (concentration 96%, sp. gr. 1835) was used instead of 12 M H_2_SO_4_ (concentration 98%, sp. gr. 1835). The sample analyses were performed in triplicate. The accuracy of the methodological procedure was also determined within our analyses by using yeast b β-glucan control sample, which is part of the analytical kit. The enclosed yeast β-glucan control sample has a known value of β-glucan content (49%). During the experiment, various appliances were used such as: laboratory scales KERN ABT 220-DM (KERN & SOHN GmbH, Balingen, Germany); vortex mixer bioSan V-1 plus, Personal Vortex (bioSan, Riga, Latvia); water bath Memmert U 1.28 (Memmert GmbH + Co. KG, Schwabach, Germany); centrifuge Hettich Mikro 185 (Andreas Hettich GmbH & Co. KG, Tuttlingen, Germany); and spectrophotometer UV/Vis Cary 60 (Agilent, Santa Clara, CA, USA).

The content of total glucans and α-glucans was evaluated separately within the established methodology. The resulting β-glucan values were determined as the difference between the values of total glucans and α-glucan. For this purpose, a specially prepared software file Mega-Calc^TM^ (Megazyme, Bray, Co. Wicklow, Ireland) was used, which was developed by the manufacturer of the analytical enzymatic kit, where the results of input data (weight of the sample, measured value of the absorbance of the sample) were recalculated and the final values of β-glucan content in dry matter in the analyzed samples were produced.

### 2.6. Statistical Analysis

The determined values were analyzed by software Statgraphics Centurion XVII (Statgraphics Technologies, Inc., The Plains, VA, USA). Analysis of variance (ANOVA) and LSD (Least Significant Difference) test with significance: nonsignificant (NS) or significant at *p* ≤ 0.05, were used. Difference was tested between the concentration of total glucan, α-glucans, and β-glucans depending on the stipe and the cap. Furthermore, the content of β-glucans in oyster mushroom fruiting bodies was statistically evaluated depending on the production strain.

## 3. Results and Discussion

### 3.1. CDDP Fingerprinting of Pleurotus ostreatus

The aim of this partial study was to analyze the possibility of differentiation of individual production strains of oyster mushroom, and/or identify possible duplication, respectively. When acquiring strains from different sources, it was not clear whether the individual collection organizations did not designate the same production strain with different designations.

As already mentioned, there is a variability between individual oyster mushroom production strains when different aspects of production potential as well as genetic polymorphism are compared.

Here, conserved DNA-derived polymorphism was evaluated among different *Pleurotus ostreatus* accessions using the primer combination F plus R2b [32]. A total of 374 amplicons were generated by PCR and distributed from 4 up to the 10 amplicons per accession with a total of 12 different length levels (Figure 3). Only one of the generated amplicon length levels was saturated by loci amplified in all of the accession, which represents 92% polymorphism (Table 2). The discrimination power for the 57 accessions reached the level of 93%.

Cophenetic correlation coefficient returned in UPGMA with a value of 0.57, which shows different types of individual CDDP amplicon profiles for the analyzed accessions of *Pleurotus ostreatus*. Polymorphic information content of the used CDDP primer pair was on average 0.45 with a range of 0.375 up to 0.5 for the individual accessions, which confirmed this marker technique to be applicable to describe the polymorphism of *Pleurotus ostreatus*.

UPGMA analysis with the Dice coefficient was used to prepare the dendrogram from the CDDP amplicons. The generated dendrogram (Figure 4) divided the evaluated *Pleurotus ostreatus* samples into two major groups of clusters with further subdivisions into a total of 19 subclusters that varied in their differences. Accessions PO-95 and PO-96, as well as accessions PO-45 and PO-89, show CDDP profiles that are identical. Based on the finding that only two pairs of identical strains were identified in the analyzed sample set, we kept them in the study focused on the glucans content determination. The purpose was to monitor them as parallel samples, confirming or denying the accuracy of glucans identification in Section 3.2. However, the results are original and are not specifically adjusted by averaging the values of each other. The possible differences in the values that we consider acceptable are probably due to the individual approach of the laboratory technician to the separation of the fruiting body into a stipe and a cap.

Conserved regions of coding parts in plant genomes are most often the functional domains that are present across plant species and thus represent an efficient source of developing DNA-based marker techniques. Different marker techniques are actually used to analyze the polymorphism generated through the coding parts of the plant genomes, such as SCoT [41], CDDP [42], TRAP [43], or PBA [44]. Besides this approach, universal or semi-universal marker techniques are still widely used to describe the variability of plant genomes, such as RAPD [45], ISSR [46], AFLP [47], or iPBS [48]. All these approaches enable the amplification of specific DNA sequences of different biological species.

CDDP was reported for plants as based on the selection of well-characterized genes involved in response to both abiotic and biotic stresses that have multiple sites in plant genomes and generate multiple polymorphic profiles [32]. It was proved to be an effective marker technique for many plant species. In 22 gerbera cultivars, all of them were differentiated and three main clusters were returned in a dendrogram of the analyzed accessions [34]. When analyzing the apple CDDP polymorphism, some primer combinations generated identical amplification profiles [35] such as in the case of our study. CDDP-based fingerprinting was reported to be applicable in the genetic diversity description as well as for germplasm conservation [49].

To our knowledge, this is the first study where CDDP markers were applied to analyze the genetic polymorphism among the *Pleurotus ostreatus* accessions as well as for fungi per se. Oyster mushroom was characterized by different DNA markers before. Genetic linkage map was prepared by RAPD and RFLP markers, est1 locus, rRNA genes, and repetitive DNA sequences [27]. A RAPD methodology was confirmed later to be suitable for the study of *Pleurotus ostreatus* genome variability [50]. CDDP resulted in polymorphic and well-reproduced fingerprints for *Pleurotus ostreatus* and is effective for its genomic variability characterization beside the traditionally used DNA markers such as RAPD or RFLP.

### 3.2. Concentration of Glucans in Production Strains of Oyster Mushroom (Pleurotus ostreatus)

Mushrooms are not only significant for their traditional edible value, but also for their medicinal value [51]. Mushrooms are known also as functional foods, mainly for their bioactive compounds that can have various beneficial impacts on human health [52]. *Pleurotus ostreatus* belongs to the important fungi, where this fungus is considered from a commercial, gastronomic, or biotechnological point of view [37]. The popularity of the oyster mushroom is due to it being well known for its relatively easy agrotechnology and relatively high content of health-promoting substances. It is often the subject of various studies. β-glucans are one of the major active components that are contained in fungi [5]. This fact is the reason why many studies are focused on the monitoring of β-glucans in oyster mushroom (*Pleurotus ostreatus*). However, in a study of the literature and articles, we found out that most of the available publications are focused on the evaluation of variability between oyster mushrooms and other fungal species. Only some publications are exclusively focused on the assessment of the variability within individual oyster mushroom production strains. This fact is not standard for the frequently grown crops. For example, with the monitoring of bioactive substances in different frequently grown fruits or vegetables, the variability of individual varieties is well processed. The present article reflects the absent data and deals with the extensive screening of glucan content in fruiting bodies of producing strains of commercial and economically important oyster mushroom.

In our research, we focused on the variability of production strains (a total of 60 strains) in terms of β-glucan content. It is common knowledge that the cap of the oyster mushroom is smooth and supple in structure, the stipe of oyster mushroom is usually stiff and tough. Therefore, often the stipe is intended for the extraction of bioactive substances in the industry, while the cap is used in gastronomy. In our study, we therefore evaluated the individual content of β-glucans in the stipe (60 strains) and in the cap (59 strains). The values are shown in the table below (Table 3). The values in the table are classified in ascending order, based on the β- glucan content in the stipe. We consider this factor to be the most important, both for the manufacturing sector that deals with the production of bioactive preparations, and for the common consumer focused on the consumption of food containing biologically active substances.

The minimum values of glucans in the cap were as follows: total glucans 24%, α-glucans 0.45%, and β-glucans 23%. The maximum values of glucans in the cap were as follows: total glucans 46%, α-glucans 3.1%, and β-glucans 45%. The minimum detected values of glucans in the stipe were as follows: total glucans 24%, α-glucans 0.31%, and β-glucans 22%. The maximum detected values of glucans in the stipe were as follows: total glucans 59%, α-glucans 6.8%, and β-glucans 56%.

In Section 3.1, we report that an identical CDDP profile was identified between samples 45 and 89 and 95 and 96. However, as it is shown in Table 3, the values of detected glucans vary, but all of them belong to the higher content for the total β-glucans (Table 3, arithmetic mean of the β-glucans content in the cap and stipe for specific strains).

The fact is that glucans have a positive effect on human health [5,54,55]. However, the priority scientific focus is on β-glucans. Mironczuk-Chodakowska et al. [56] state that β-glucans are natural molecules that have beneficial potential thanks to their immunomodulatory, anti-inflammatory, antineoplastic, antibacterial, antifungal, antioxidant, and anti-allergic antiviral properties. The concentration of β-glucans is several times higher compared to α-glucan. According to our observations, it is clear that the stipe contained on average 2081% and the cap 2471% more β-glucans than α-glucans. The scientific studies attribute healing effects both to α-glucans as well as to β-glucans. However, from the point of view of practice, due to a low concentration of α-glucans in the fruiting body, they are not very interesting [57,58]. The results of our measurements are shown in Table 4.

The measured and processed data show that different production strains of *Pleurotus ostreatus* have different glucan contents in the different parts of the fungus. The results of the experiment clearly indicate a higher content of glucans in the stipe compared to the cap. In the case of α-glucan, a difference in their concentration in the stipes versus the cap was detected, with 57% in favour of the stipe. In the case of β-glucans, the conclusion was the same, the higher concentration of β-glucans was also in the stipe, specifically by 33%, compared to the concentration of β-glucans in the cap. All differences are statistically significantly different at the 95.0% confidence level. The average values of total glucans, α-glucans, and β-glucans in the stipe and in the cap in the analyzed samples of different production strains of *Pleurotus ostreatus* are shown in Table 4. The total values of β-glucans content were processed further and divided into the 5 different levels of the content from the minimal value of 29.8 up to the maximum of 47.7 (the arithmetic mean of the β-glucans content of cap and stipe for each specific strains is shown in a separate column in Table 3) with the division factor 3.5, and the PCoA analysis was performed based on the Dice coefficient (Figure 5). Four groups were generated on the levels of object scores.

The topological specificity of glucan content in the fruiting bodies was also proved in the study from Synytsya et al. [59] and it stated that stipes contain more β-glucans than caps. In the published results from the study by Synytsya et al. [59], it was stated that the content of α-glucan in the samples of *Pleurotus ostreatus* was in the range of 3.4 to 7.9% in the caps and of 3.0 to 7.6% in the stipes. Compared to our results, these authors report a 303% higher α-glucan content in the cap and a 141% higher α-glucan concentration in the stipe. From our point of view, this variability can be caused by several reasons. The decisive factor is the variability caused by using other agricultural techniques in the cultivation of biological material and substrate, as well as the selection of the production strain, which we clearly demonstrate in our study. According to the same authors, the content of β-glucan in the in the samples of *P. ostreatus* was as follows: 27.4–39.2% in the caps and 35.5–50.0% in the stipes. Our findings suggest a 9% higher β-glucans content in the cap and a 12% higher in the stipe. In this case, we achieved almost identical results. The difference in the glucan content was not only observed in *Pleurotus ostreatus*, but also in other species of mushroom. In the study from Sari et al. [60], it was observed that in most of the analyzed samples of mushroom species, which were divided into the stipes and the caps, there was a higher β-glucan content in their stipes then in the caps. The same difference of β-glucan content in the stipes and in the caps was also shown among wild grown mushrooms, e.g., in *Boletus edulis* there was determined to be a big difference between β-glucan content in the cap (17%) and in the stipe (58%) [60]. In the experiment of Vetvicka et al. [17], in *Pleurotus eryngii* a slightly different content of α-glucans in the cap of mushroom was detected. The content was 0.8% in comparison with our result of 1.2%. In the stipe they traced a much higher content of α-glucans of 4.5%. Our content was 2.2%. As for β-glucans, quite comparable content was found in both cap and stipe. In cap they detected 29.5% of β-glucans; in our experiment it was 36%. The content of β-glucans in stipe according to their results was 38.4%, in ours it was 48%. However, we would like to point out that the study does not reflect strain variability, but only mentions average values. β-glucans, as an important bioactive compound in human nutrition, can be extracted from fruiting bodies in various ways. The extraction is most often carried out with hot water [61]. From the producers’ points of view, the decisive aspect of the profitability of the extraction is the amount of β-glucans in the fruiting bodies of oyster mushroom. It is therefore important for this process to select production strains characterized by their high content.

In the study, we did not perform a statistical analysis of glucan content between strains on purpose. The reason is that the result could be distorting and misleading. Data variability in individual glucan content is large and the differences between strains are low (Table 3). The analyses were performed accurately, and the results are objective. However, in the case of *Pleurotus ostreatus*, it is not possible to separate the fruiting body into a stipe and a cap as precisely as, for example, in the species *Agaricus bisporus*. We therefore consider it confusing to present the results through statistics.

In our study, we detected the lowest content of β-glucans (in the stipe) in strain no. 72 and the highest in strain no. 43. The variability ranged between 22% and 56%, which is a difference in the concentration of 34%. As we did not find a relevant source in the scientific literature that characterizes strains with low, medium, and high content, we divided our results as follows. In order to quickly orient the reader in the data, we divided the data evenly into three groups (Table 5).

We used this division in the assessment of production strains (Table 3) and it might be used for material for further comparative studies by other authors. The work did not consider the yield potential of individual production strains. The fertility parameter of individual strains may be significant, as there is evidence that the biological efficiency of individual strains may vary [62,63,64]. *Pleurotus ostreatus* species is quite unique because they are able to grow on different agrowaste materials [65], such as rice straw [66], oil palm fronds [67], empty fruit bunch and palm-pressed fiber [68], sugarcane bagasse [69], corn cob [70] wheat straw, coffee waste, and leaf litter of forest trees [71]. If the strain is characterized by a high content of β-glucans but is low in crop, it is not an important source of biologically valuable substances for the producer. In this case, we recommend choosing a compromise between β-glucan content and fertile potential.

When comparing the results of total average β-glucan content in the analyzed strains and generated CDDP profiles (Figure 6), the correlation coefficient reached the value of 0.12, which presents no specific correlation for the whole group of analyzed samples. When regarding some specific groups of strains, the correlation was stronger. For the group of low content of β-glucan (from 29.8 up to the 36.8), the negative correlation with the generated CDDP profiles haf the value −0.34, and in this group, less CDDP fragments were amplified. Medium positive correlation with a value of 0.58 was found in the small group of strains that were identical in the CDDP profiles (PO-45/PO-89 and PO-95/PO-96), where all of them belong to the strains with higher content of total average β-glucans), but for the whole group of strains with higher content this was not repeated.

## 4. Conclusions

The work confirmed our hypotheses that, as with other agricultural species of vegetables and fruits, there is variability in the content of selected bioactive substances between the production strains of important edible and medicinal mushrooms. Significant differences were found between the content of β-glucans in the individual production strains with each other as well as in the stipe versus in the cap.

Based on the presented study, it is possible to select from 60 production strains of *Pleurotus ostreatus* those that are characterized by a high content synthesis of β-glucan and use them as a genetic material for the production of functional preparations usable in human nutrition. At the same time, utilization of these strains for the production of fruiting bodies used in gastronomy is possible to directly increase the nutritional value of the food prepared from these fruiting bodies. However, from the point of view of production profitability, we recommend further studies paying attention to the fertility potential of the set strains, which was not set as a research goal in this study.

Here, the CDDP marker technique was applied for the first time to analyze the genetic polymorphism of *Pleurotus ostreatus* and was proved as an effective tool in the discrimination of the production strains of this popular edible mushroom. Information generated by DNA-based markers are useful in the management of genetic resources as well as in the breeding of them.

## Figures and Tables

**Figure 1 jof-08-00563-f001:**
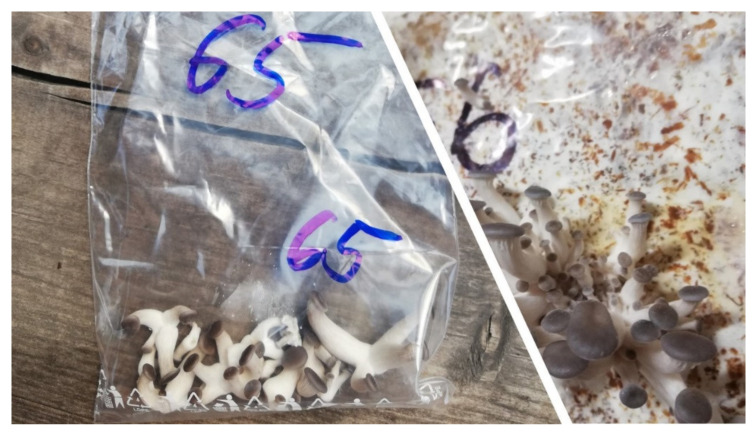
Young fruiting bodies of oyster mushroom in the harvesting stage (source: author of the work).

**Figure 2 jof-08-00563-f002:**
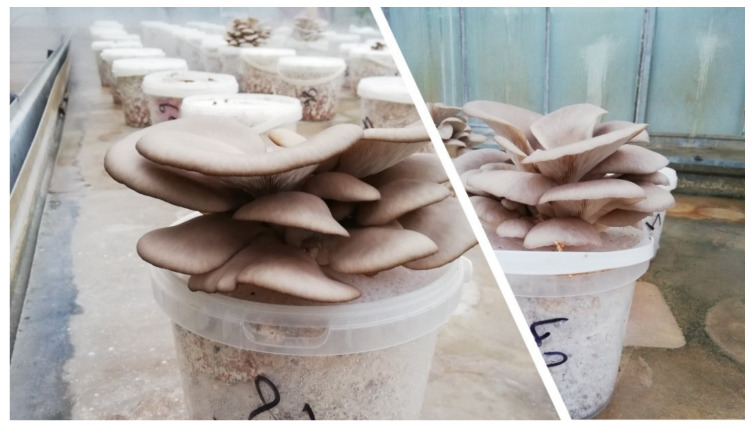
Fruiting bodies of oyster mushroom in the harvesting stage (source: author of the work).

**Figure 3 jof-08-00563-f003:**
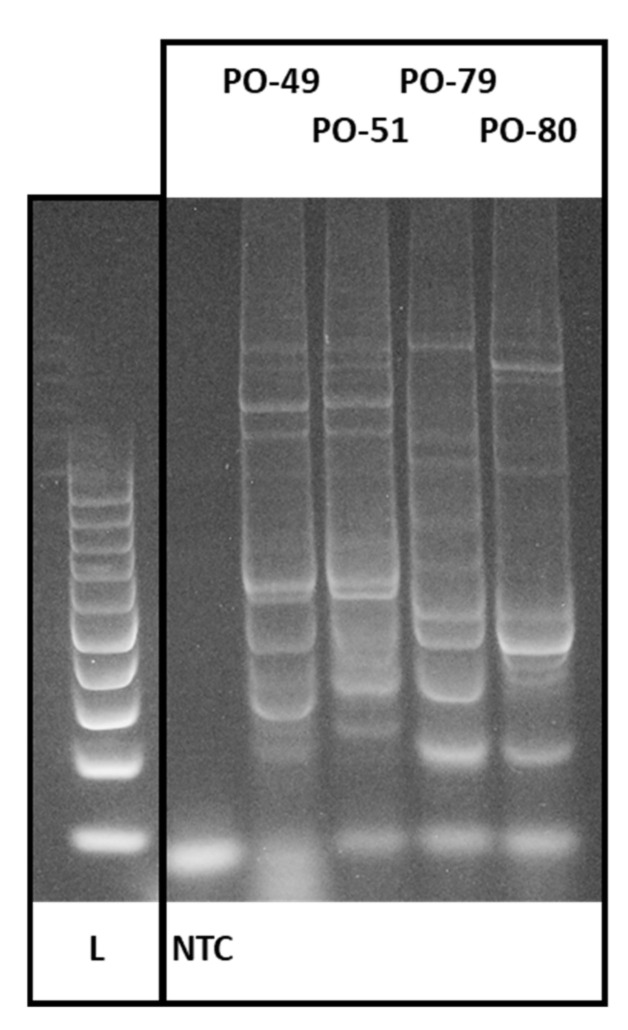
CDDP profiles of selected oyster mushroom production strains analyzed in this study. L—100 bp ladder; NTC—non template control (Source: Author of the work).

**Figure 4 jof-08-00563-f004:**
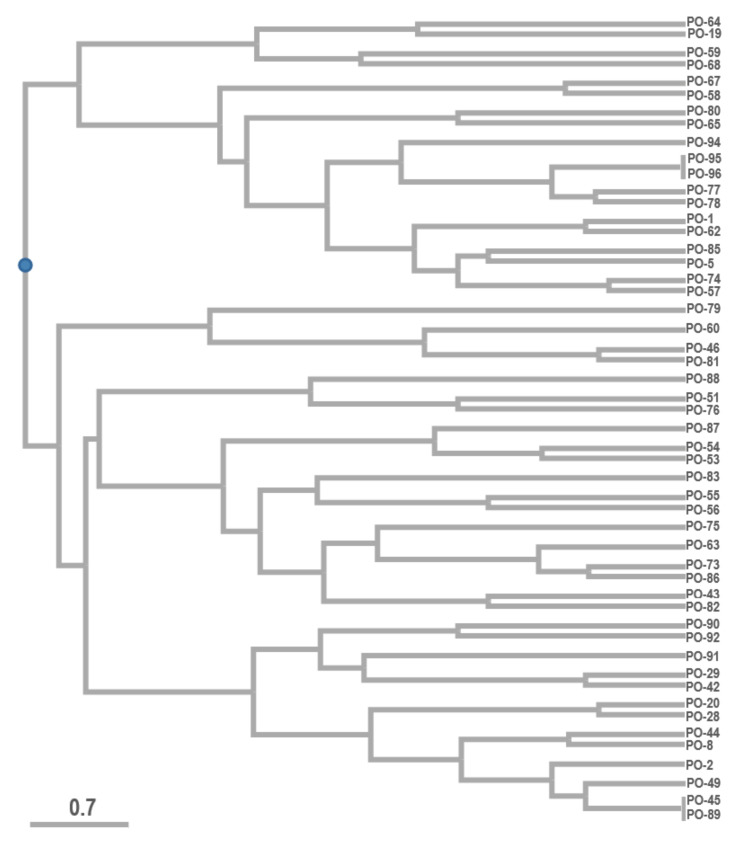
UPGMA-based dendrogram of analyzed *Pleurotus ostreatus* accessions evaluated by CDDP markers. Note: PO-XY—*Pleurotus ostreatus* and numbers of production strains (Source: Author of the work).

**Figure 5 jof-08-00563-f005:**
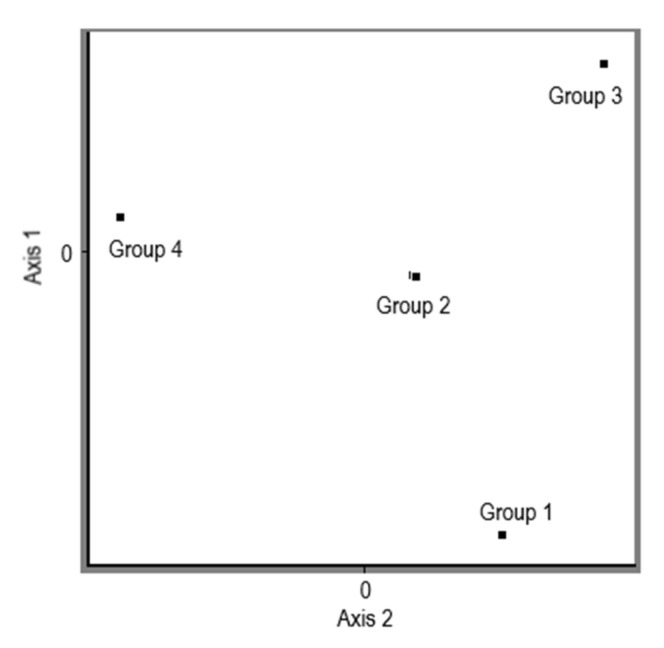
PCoA analysis of the range of total β-glucans in the analyzed stipes of *Pleurotus ostreatus*. Group 1: PO-1, PO-5, PO-20, PO-48, PO-54, PO-55, PO-64, PO-70, PO-71, PO-77, PO-78, PO-87, PO-94, and PO-102; Group 2: PO- 2 and PO-49; Group 3: PO-8, PO-19, PO-29, PO-43, PO-44, PO-51, PO-60, PO-61, PO-67, PO-74, PO-75, PO79, PO-81, PO-83, PO-92, PO-96, and PO-97; Group 4: PO-28, PO-42, PO-45, PO46, PO-56, PO-57, PO-58, PO-59, PO-62, PO-65, PO-68, PO-69, PO-73, PO-76, PO-80, PO82, PO-84, PO-85, PO-86, PO-88, PO-89, PO-90, PO-91, PO-95, and PO-128.

**Figure 6 jof-08-00563-f006:**
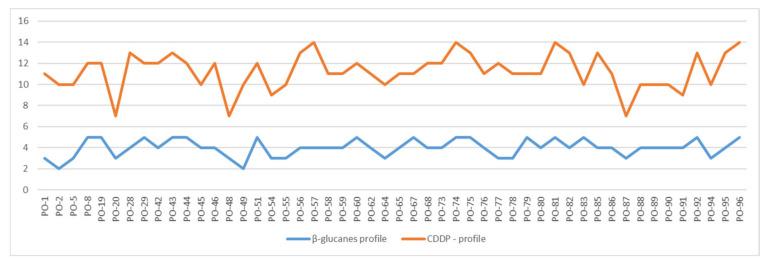
Profiles of correlated β-glucan content and CDDP profiles of analyzed strains of *Pleurotus ostreatus*.

**Table 1 jof-08-00563-t001:** Biological material.

Number	Designation in the Collection	Identification	Further Identification and Description
1		*Pleurotus ostreatus* HK35	dr. Jablonský, Czech University of Life Sciences Prague, Czech Republic
2		*Pleurotus ostreatus* Kryos B	dr. Jablonský, Czech University of Life Sciences Prague, Czech Republic
5		*Pleurotus ostreatus* P-80	Mr. Rajtár, Mycoforest Company, Slovakia,
8		*Pleurotus ostreatus*	dr. Pavlík, Zvolen, spruce harvest, Slovakia
19		*Pleurotus ostreatus* 2175	Mr. Rajtár, Mycoforest Company, Slovakia
20		*Pleurotus ostreatus* CHINA BLACK	Mr. Rajtár, Mycoforest Company, Slovakia
PL-28		*Pleurotus ostreatus* PL-28	commercial strain
28		*Pleurotus osteratus*	isolate from the market, Slovakia
29		*Pleurotus osteratus*	origin unknown
42		*Pleurotus osteratus* MEY 2191	Mr. Rajtár, Mycoforest Company, Slovakia
43		*Pleurotus ostreatus* GIZA	Mr. Rajtár, Mycoforest Company, Slovakia
44		*Pleurotus ostreatus* K12	Mr. Rajtár, Mycoforest Company, Slovakia
45		*Pleurotus ostreatus* RH	Mr. Rajtár, Mycoforest Company, Slovakia
46		*Pleurotus ostreatus* K6	Mr. Rajtár, Mycoforest Company, Slovakia
48		*Pleurotus osteratus*	origin unknown
49		*Pleurotus osteratus*	origin unknown
51	CPPF-5001	*Pleurotus ostreatus*	Crop Research Institute, Czech Republic
53	CPPF-5002	*Pleurotus ostreatus* Pl2	Crop Research Institute, Czech Republic
54	CPPF-5019	*Pleurotus ostreatus* P-84	Crop Research Institute, Czech Republic
55	CPPF-5022	*Pleurotus osteratus*	Crop Research Institute, Czech Republic
56	CPPF-5075	*Pleurotus osteratus*	Crop Research Institute, Czech Republic
57	CPPF-5117	*Pleurotus osteratus*	Chna 4, Crop Research Institute, Czech Republic
58	CPPF-5141	*Pleurotus osteratus*	PO-DD-1/1, Crop Research Institute, Czech Republic
59	CPPF-5142	*Pleurotus osteratus*	PO-SV-1/1, Crop Research Institute, Czech Republic
60	CPPF-5143	*Pleurotus osteratus*	PO-PH-1/1A, Crop Research Institute, Czech Republic
61 *	CPPF-5144	*Pleurotus osteratus*	PO-HOR-1/2, Crop Research Institute, Czech Republic
62	CPPF-5145	*Pleurotus osteratus*	PO-HOR-2/4, Crop Research Institute, Czech Republic
63	CPPF-5146	*Pleurotus osteratus*	PO-CB-1/2, Crop Research Institute, Czech Republic
64	CPPF-5147	*Pleurotus osteratus*	PO-HD-1/1A, Crop Research Institute, Czech Republic
65	CPPF-5148	*Pleurotus osteratus*	PO-HD-2/1, Crop Research Institute, Czech Republic
66	CPPF-5149	*Pleurotus osteratus*	PO-LZ-1/1, Crop Research Institute, Czech Republic
67	CPPF-5150	*Pleurotus osteratus*	PO-MV-1/1A, Crop Research Institute, Czech Republic
68	CPPF-5151	*Pleurotus osteratus*	PO-SK-1, Crop Research Institute, Czech Republic
69 *	CPPF-5153	*Pleurotus osteratus*	PO-SK-3, Crop Research Institute, Czech Republic
70 *	CPPF-5155	*Pleurotus osteratus*	PO-SK-5, Crop Research Institute, Czech Republic
71 *	CPPF-5156	*Pleurotus osteratus*	PO-PSB, Crop Research Institute, Czech Republic
72 *	CPPF-5173	*Pleurotus osteratus*	Po-OH--JR1A, Crop Research Institute, Czech Republic
73	CPPF-5177	*Pleurotus osteratus*	PO ŠMA, Crop Research Institute, Czech Republic
74	CPPF-5179	*Pleurotus osteratus*	Crop Research Institute, Czech Republic
75	CPPF-5192	*Pleurotus osteratus*	from Hlíva Huť, Crop Research Institute, Czech Republic
76	CEMM012 (VURV-M12)	*Pleurotus osteratus*	210-ENV, dr. Havránek, 2009, Olomouc, Crop Research Institute, Czech Republic
77	CEMM013 (VURV-M13)	*Pleurotus osteratus*	93-PLV, dr. Havranek 2008, Pohořany, Crop Research Institute, Czech Republic
78	CEMM118 (VURV-M118)	*Pleurotus osteratus*	PLM pl, dr. Petrželová 2016, PR Doubrava (from Mora-vičany-Bradlec), Crop Research Institute, Czech Republic
79	CEMM119 (VURV-M119)	*Pleurotus osteratus*	PLNZ sp1, dr. Petrželová, 2016, CHKO Litovelské Pomoraví (from Nové Zámky a Nový Dvůr), Crop Research Institute, Czech Republic
80	CEMM120 (VURV-M120)	*Pleurotus osteratus*	PLO sp, Dr. Egertová, Sochor 2015, Olomoučany, Crop Research Institute, Czech Republic
81	CEMM121 (VURV-M121)	*Pleurotus osteratus*	PLP pl, dr. Jurková, 2013, Pohořany, Crop Research Institute, Czech Republic
82	CCBAS 278	*Pleurotus osteratus*	dr. Semerdžieva, 1993, Crop Research Institute, Czech Republic
83	CCBAS 459	*Pleurotus osteratus*	G. Ritter 1956, Schierke, Harz mountains, Germany, Crop Research Institute, Czech Republic
84 *	CCBAS 462	*Pleurotus osteratus*	E. Jones 1966, England, Great Britain, Crop Research Institute, Czech Republic
85	CCBAS 472	*Pleurotus osteratus*	W. Luthart 1959, České Budějovice, Crop Research Institute, Czech Republic
86	CCBAS 473	*Pleurotus osteratus*	W. Luthart 1960, České Budějovice, Crop Research Institute, Czech Republic
87	CCBAS 474	*Pleurotus osteratus*	dr. A. Torev, 1965, Plovdiv, Bulgaria, Crop Research Institute, Czech Republic
88	CCBAS 476	*Pleurotus osteratus*	dr. A. Ginterová, 1973, Svatý Jur near Bratislava, Slovakia, Crop Research Institute, Czech Republic
89	CCBAS 684	*Pleurotus osteratus*	dr. M. Semerdžieva, 1983, Gaštanica near Nitra, Slovakia, Crop Research Institute, Czech Republic
90	CCBAS 692	*Pleurotus osteratus*	dr. I. Ohira, 1975, Shuzenzi-cho, Pref. Shizuoka, Japan, Crop Research Institute, Czech Republic
91	CCBAS 757	*Pleurotus osteratus*	dr. M. Semerdžieva, 1987, near Trutnov, Crop Research Institute, Czech Republic
92	CCBAS 766	*Pleurotus osteratus*	dr. M. Semerdžieva, 1985, Crop Research Institute, Czech Republic
94		*Pleurotus osteratus*	isolate from the market, Slovakia, 2019, SPOREA, origin Poland
95		*Pleurotus osteratus*	isolate from the market, Slovakia, 2019, origin Slovakia
96		*Pleurotus osteratus*	isolate from the market, Slovakia, 2019, České houby, Czech Republic
97 *		*Pleurotus osteratus*	origin Czech Republic
102 *		*Pleurotus osteratus*	isolate from the market, Slovakia, 2019, České houby, from ČR, Czech Republic

Source: Author of the work, also in Golian et al. [37] * marked strains were not a part of CDDP analysis.

**Table 2 jof-08-00563-t002:** CDDP loci characteristics in the analyzed oyster mushroom accessions.

Locus Number	Lf	Ne	H	I
1	Allele 0: 0.4118	1.9396	0.4844	0.6775
	Allele 1: 0.5882			
2	Allele 0: 0.5490	1.9810	0.4952	0.6883
	Allele 1: 0.4510			
3	Allele 0: 0.5882	1.9396	0.4844	0.6775
	Allele 1: 0.4118			
4	Allele 0: 0.4706	1.9931	0.4983	0.6914
	Allele 1: 0.5294			
5	Allele 0: 0.4510	1.9810	0.4952	0.6883
	Allele 1: 0.5490			
6	Allele 0: 0.4902	1.9992	0.4998	0.6930
	Allele 1: 0.5098			
7	Allele 0: 0.0	1.0	0.0	0.0
	Allele 1: 1.0			
8	Allele 0: 0.5490	1.9810	0.4952	0.6883
	Allele 1: 0.4510			
9	Allele 0: 0.2353	1.5622	0.3599	0.5456
	Allele 1: 0.7647			
10	Allele 0: 0.1176	1.2620	0.2076	0.3622
	Allele 1: 0.8824			
11	Allele 0: 0.4314	1.9630	0.4906	0.6837
	Allele 1: 0.5686			
12	Allele 0: 0.3922	1.9111	0.4767	0.6697
	Allele 1: 0.6078			

Note: Lf—allele frequency; Ne—effective number of alleles; H—Nei’s gene diversity; I—Shannon’s information index (Source: Author of the work).

**Table 3 jof-08-00563-t003:** The average glucans (%) content in the stipe and the cap in the individual production strains of *Pleurotus ostreatus*.

Production Strain of *Pleurotus ostreatus*	TG				AG				BG					Evaluation Considering β-Glucans Content
no.	Cap	±SD	Stipe	±SD	Cap	±SD	Stipe	±SD	Cap	±SD	Stipe	±SD	average value	Concentration *
43	40	3.3	58	4	1	0.49	2	0.38	39	3.8	56	3.9	48	high
44	42 *	7.3	54 *	7.5	0.9	0.088	0.6	0.061	41	7.2	54	7	48	high
74	37	3.1	58	2.5	1.9	0.11	2.9	0.49	35	1.1	55	2.6	45	high
81	39	3.6	58	6.2	3	0.53	2.8	0.063	36	7.3	55	9.5	46	high
61	42	3.7	53	5.8	1.7	0.2	1	0.083	40	5.1	52	5.7	46	high
67	42	2.9	57	4.8	2.4	0.51	5.1	0.75	39	5.3	52	4.2	46	high
79	42	6.4	52	8.1	1.6	0.35	1.6	0.36	41	8.2	51	9.4	46	high
97	46	6.2	50	6.3	0.7	0.11	1.6	0.18	45	6.3	48	7.8	47	high
8	37	2.9	54	5.6	1.2	0.27	1.8	0.31	36	2.6	52	7.7	44	high
96 (** 95)	39	3.3	54	3.0	1.5	0.16	1.8	0.21	37	3.2	52	2	45	high
19	40	5.1	55	5.9	1.5	0.14	3.4	0.054	38	5	51	5.9	45	high
66	n.d.	n.d.	51	4.4	n.d.	n.d.	0.8	0.012	n.d.	n.d.	51	4.9	n.d.	high
88	36	7.4	52	7.4	0.5	0.012	0.6	0.067	36	7.2	51	12	44	high
92	38	2.5	52	3.9	0.5	0.028	0.5	0.059	38	2.5	51	3.9	45	high
28	40	6.5	52	5.9	1.5	0.24	3.1	0.47	38	6.1	49	6.2	44	high
69	40	3.3	56	4.3	0.7	0.065	6.8	0.65	39	3.2	49	4.3	44	high
83	42	4.4	51	6.9	1.2	0.24	1.4	0.11	41	5.3	49	5.9	45	high
29	42	4.9	50	3.2	1	0.22	1.4	0.17	41	4.2	48	4.2	45	high
90	40	2.3	49	3.1	0.8	0.11	1	0.14	39	2.8	48	3.2	44	high
46	40	7.7	48	7.3	1.4	0.17	1.2	0.13	39	4.6	47	7.6	43	high
51	46 *	6.4	50 *	5.4	2.8	0.37	3	0.33	43	4.3	47	5.2	45	high
59	41	3.2	48	4.7	1	0.17	1	0.14	40	2.9	47	4.1	44	high
60	42	5.4	50	5.5	1	0.19	3.1	0.12	41	3.6	47	4.9	44	high
75	46	4.9	50	5.1	3.1	0.47	4.4	0.34	43	5.5	46	5.7	45	high
76	44	4.7	47	4.2	2.7	0.46	2.2	0.42	42	1.7	44	7.5	43	medium
89 (** 45)	44	6.2	45	6.6	1	0.28	1	0.26	43	4.1	44	7	44	medium
56	34 *	2.7	56 *	5.9	1.8	0.16	3.3	0.39	33	2.5	52	5.5	43	high
86	36	7.4	55	6.2	1.6	0.15	4	0.32	34	7.2	51	10	43	high
45 (** 89)	36	3.3	53	3.5	1.5	0.22	4.2	0.33	35	4.2	49	3.9	42	high
58	37	4.6	50	5.1	1.5	0.23	1.7	0.21	35	5.1	49	5.7	42	high
62	38	2.7	51	3.4	2	0.29	2	0.34	36	3.6	49	2.2	43	high
PL-28	37	7.5	50	5.3	1.1	0.32	0.7	0.077	36	3.4	49	6.4	43	high
102	35	6.2	51	5.6	1.9	0.42	3.3	0.36	33	4.1	48	5.7	41	high
57	37	4.2	52	5.4	1.7	0.34	3.8	0.55	35	5.3	48	7.7	42	high
80	35	2.3	51	6.5	1.6	0.31	2.7	0.19	33	1.8	48	6.5	41	high
94	34	4.6	50	4.9	1.7	0.13	2.2	0.19	32	3.3	48	5.5	40	high
73	38	3.6	51	4.9	2.4	0.56	4.2	0.41	36	3.6	47	4.2	42	high
82	38	5.8	49	6.6	1.3	0.41	2.3	0.25	37	5.6	47	6.8	42	high
65	38	6.3	48	5.7	1.4	0.37	2.2	0.31	37	3.5	46	5.2	42	high
85	38	5.9	49	5.5	2.1	0.68	3.6	0.28	36	5.2	46	5.6	41	high
91	39	3.0	46	3.3	0.8	0.11	0.9	0.012	38	4.2	46	5.1	42	high
95 (** 96)	37	7	47	6.7	1.3	0.22	1.5	0.16	36	4.3	46	6.0	41	high
64	34	0.93	46	2.4	0.6	0.027	0.7	0.035	34	0.9	45	5.3	40	high
42	41	5.8	46	3.6	1.6	0.23	2	0.11	39	4.5	44	3.6	42	medium
78	35	4.6	45	5.2	0.6	0.074	1	0.19	35	5.4	44	4.1	40	medium
84	40	7.7	44	6.8	0.4	0.037	0.3	0.022	40	7.7	44	6.8	42	medium
70	37	5.4	42	6.5	0.7	0.078	0.5	0.083	37	5.3	41	5.4	39	medium
68	42	2.5	42	3.4	0.5	0.087	1.5	0.14	42	3.8	40	3.8	41	medium
1	27 *	1.2	55 *	7.1	1.3	0.067	3.4	0.47	25	1.2	52	7	39	high
5	30 *	2.7	51 *	6.4	0.7	0.062	2.2	0.34	29	1.9	48	6.7	39	high
55	32 *	3.6	51 *	4.6	1.3	0.11	3.8	0.29	31	3.4	47	6.1	39	high
87	34	4.4	48	5.2	1.5	0.14	3.4	0.52	33	1.7	45	5.2	39	medium
20	29 *	2.9	51 *	4.6	1	0.16	3.1	0.15	28	3.3	48	4.7	38	high
48	31 *	2.7	50 *	7.9	1	0.34	2.7	0.37	30	1.9	47	7.3	39	high
54	33	5.9	47	4.9	1.1	0.061	2.5	0.55	31	3.4	44	5.3	38	medium
71	35	4.6	42	5.8	0.6	0.13	0.7	0.036	35	4.5	41	5.5	38	medium
77	36	3.4	43	3.9	1.9	0.34	1.9	0.39	34	3.2	41	3.8	38	medium
49	29	4.8	46	4.1	0.6	0.062	2.1	0.34	28	4.8	44	3.9	36	medium
2	24 *	1.9	47 *	3.8	1.2	0.11	2.8	0.29	23	2.1	44	4.2	34	medium
72	38	6.2	24	5.6	1	0.19	1.1	0.11	37	2.1	22	2.8	30	low

Note: no—number, TG—total glucan, AG—α-glucan, BG—β-glucan, ±SD = standard deviation, n.d.—** according to the CDDP profile also strain number XY. Source: Author of the work, * also in Chlebová et al. [53].

**Table 4 jof-08-00563-t004:** The average values of total glucans, α-glucans, and β-glucans in the stipe and cap in the analyzed samples of *Pleurotus ostreatus*.

		Total Glucans (%)		α-Glucans (%)		β-Glucans (%)	
Variant	*n*	Average	±SD	Average	±SD	Average	±SD
Cap	59	38 ^a^	4.6	1.4 ^a^	0.63	36 ^a^	4.4
Stipe	60	50 ^b^	5.2	2.2 ^b^	1.3	48 ^b^	4.8

Note: ±SD = standard deviation, *n*—number of production strains in the analysis; values with different lowercase letters in column are significantly different at *p* < 0.05 by LSD in ANOVA. Source: author of the work.

**Table 5 jof-08-00563-t005:** The distribution of strains based on adequate β-glucan content in the fruiting body.

Level	Content of β-Glucans in %
Low	from 22 to 33
Medium	from 34 to 45
High	from 46 to 56

Source: Author of the work.

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
