# Peer review of "Analysis of Biochemical and Genetic Variability of Pleurotus ostreatus Based on the β-Glucans and CDDP Markers"

_jof, 2022, doi:10.3390/jof8060563_

Round 1

Reviewer 1 Report

Dear authors,

After a revision for the article entitled "Analysis of biochemical and genetic variability of Pleurotus ostreatus based on the β-glucans and CDDP markers" under the journal number jof-1701829

I give a moderately favorable opinion after major corrections.
- It is necessary to specify many details in the material and method part
- it is necessary to make the link between the parties
-Even with the biochemical part you can illustrate your mushrooms in a tree or an ACP.
- the genetic part is very poorly done (lacks a lot of details)
- lack of balance between the two parties which devalues the content of the article.
-you can improve your statistical analyzes for both parts of the job.
if the article will be corrected well could be accepted in a journal with high impact factor like JOF considered for the original results

Good luck

Author Response

Dear reviewer,

Thank you very much for your comments, which after incorporation have definitely improved the quality of our manuscript. Below we answer to specific comments in points.

--- it is useless to put this information in the abstract

///Thank you for your remark, but according to MDPI instructions the exact method of determination must be given in the abstract. Therefore, we have retained this phrase.

--- It is necessary to specify many details in the material and method part

--- it is necessary to put more detail on the genetic part the content of the abstract is not very informative on the content of the article

--- you are missing at the end a small paragraph describing the detailed work objective

--- mL

--- your other comments

/// Thank you, we tried to make corrections and additions.

--- for all the primers tested the hybridization temperature is 45°C????

///Hybridization temperature of selected primer pair was 54°C, as is written in the manuscript. 45 seconds was the time for primer hybridization.

--- it is necessary to make the link between the parties,

--- Even with the biochemical part you can illustrate your mushrooms in a tree or an ACP.

--- the genetic part is very poorly done (lacks a lot of details)

 ///Dear reviewer,

Thank you for reviewing our manuscript and for your comment. We tried to create a connection between the individual chapters, changing the order of chapters in the manuscript.

The problem arose as follows. The study was developed by two independent research teams. The first solved the primary task (completely rational), to identify the same strains of oyster mushroom and thus reduce the number of samples and thus the financial and time difficulty of the experiment. The biological material was from different sources and under the different designations, we did not know which samples were parallel. However, the genetics team had to optimize the entire methodology to meet this goal. After the successful implementation of the experiment, we considered it appropriate to publish it together, originally with the main part - the analysis of glucans in the fruiting bodies of oyster mushroom. The genetics team found almost complete authenticity of the oyster mushroom strains, as the parallel samples were evaluated only 4 of all the analyzed samples. Therefore, we decided to keep all the strains in order.

The second research team was focused on the main part of our study, and that the identification of glucans in 60 production strains, individually in the stipes and in the caps. To our knowledge, such an extensive study is unparalleled.

Changes made: change of chapters' order; addition of explanatory and connecting phrases; renumbered tables, chapters, and literature; strengthening and complementing the genetic part

--- lack of balance between the two parties which devalues the content of the article.

///Dear reviewer, we explain the shortness of the chapter in the text above. This is only a primary study. However, we have tried to strengthen the chapter.

--- you can improve your statistical analyzes for both parts of the job

///Dear reviewer,

In the part dealing with glucan analysis (Table 2), a column was added in the original manuscript dividing all strains into homogeneous groups at Method: 95.0 percent LSD. Based on the great variability of the data, 9 homogeneous groups were created. It is possible, but not meaningful, to present these results. After careful consideration, together with the colleagues from the research team we decided before sending the manuscript that we would not use statistics when evaluating strains. If we want to present the difference in each other in production strains, for example in the content of β-glucans, it would be necessary to eliminate possible errors by certainty, for example by subjectively separating the stipe from the cap. We suppose that our results are correct and objective, but with such a low difference in the glucan content between strains, we do not consider it correct to present them using statistics, given the small errors in manual performance. In case of your interest, we can send you the values immediately.

Another case is the evaluation of the glucan content in the stipe vs. in the cap. Here the results are so different that even after taking into account any small errors of the manipulator, we consider them highly objective and therefore we present them in the work.

The following paragraph was added to the text of the manuscript: In the study, we did not perform a statistical analysis of glucan content between strains on purpose. The reason is that the result could be distorting and misleading. Data variability in individual glucan content is large and the differences between strains are low (Table 2). The analyzes were performed accurately and the results are objective. However, in the case of Pleurotus ostreatus, it is not possible to separate the fruiting body into a stipe and a cap as precisely as, for example, in the species Agaricus bisporus. We therefore consider it confusing to present the results through statistics.

We firmly believe that we have fulfilled your expectations by editing the manuscript.

Sincerely, the authors of the work.

Reviewer 2 Report

Dear authors

There are two major sections the one that is about variability in glucan content and the other that there is genetic variability in the different strains . For these  two parts there  is not evidence how  they are related. How marker varibility is related to glucan and also proven with statistical analysis. In the data there is not one gel photo  for the marker analysis.  It is not mentioned from the gel photos to digitization if software was used or was made manually. There is definetly enough work in culture of the fungi and the glucan  estimation but for the target of this paper is not  enough. It needs a lot of meta data analysis about the genetic sequences (which are not provided ) the strains to which they relate  and if the dendrogam has any relation with the  glucan content.  At this stage it seems as two papers put together and are missing the connecting information.

Author Response

Dear reviewer,

Thank you very much for your comments, which after incorporation have definitely improved the quality of our manuscript. Below we answer to specific comments in points.

---  For these two parts there is no evidence how they are related.

///Dear reviewer,

Thank you for reviewing our manuscript and for your comment. We tried to create a connection between the individual chapters, changing the order of chapters in the manuscript.

The problem arose as follows. The study was developed by two independent research teams. The first solved the primary task (completely rational), to identify the same strains of oyster mushroom and thus reduce the number of samples and thus the financial and time difficulty of the experiment. The biological material was from different sources and under the different designations, we did not know which samples were parallel. However, the genetics team had to optimize the entire methodology to meet this goal. After the successful implementation of the experiment, we considered it appropriate to publish it together, originally with the main part - the analysis of glucans in the fruiting bodies of oyster mushroom. The genetics team found almost complete authenticity of the oyster mushroom strains, as the parallel samples were evaluated only 4 of all the analyzed samples. Therefore, we decided to keep all the strains in order.

The second research team was focused on the main part of our study, and that the identification of glucans in 59 production strains, individually in the stipes and in the caps. To our knowledge, such an extensive study is unparalleled.

The absent connection between the two individual chapters in the manuscript argues that one team did not want to interfere in the work of the other team. However, we have made a correction.

Changes made: change of chapters' order; addition of explanatory and connecting phrases; renumbered tables, chapters, and literature; strengthening and complementing the genetic part

--- How marker varibility is related to glucan and also proven with statistical analysis. In the data there is not one gel photo  for the marker analysis.  It is not mentioned from the gel photos to digitization if software was used or was made manually.

///We decided not to add gel photo to the manuscript, as many others and according to our opinion more relevant figures are presented there.

--- There is definetly enough work in culture of the fungi and the glucan  estimation but for the target of this paper is not  enough.

/// Dear reviewer, we appreciate your recognition. We are glad that you can really appreciate all the work that has been done. The data of the glucan content itself is unique and there are no alternatives in the available studies. We understand that for the JoF the presented data might not appear to be sufficient, so we tried to strengthen the genetic part of the study. We would also like to draw your attention to the fact that this is an ongoing study of the production strains. In the January issue of journal Foods, we published a manuscript focused on identifying the mineral composition of these production strains. https://doi.org/10.3390/foods11010076

–-It needs a lot of meta data analysis about the genetic sequences (which are not provided ) the strains to which they relate  and if the dendrogam has any relation with the  glucan content.  At this stage it seems as two papers put together and are missing the connecting information.

///New informations were added and the connection points are discussed.

Reviewer 3 Report

The manuscript presents the glucan content and the genetic diversity of a group of Pleurotus ostreatus strains, which is essential for utilizing and selecting the germplasm of mushrooms. However the two parts of the results are lack of association, which still needs to be comprehensisvely analyzed and discussed. Specifically,

Line35: a number of

Line 37 which are nontoxic and biodegradable

Line60-62: “To this day, have been conducted…”, Please rephrase the sentence.

Lines: What do you mean by “oyster mushroom production strains”, are they commercial cultivars or strains can produce mushrooms? If you meant commercial cultivars, I doubt you might mix the strains of different species of Pleurotus mushrooms. Base on the fact that the taxonomy of Pleurotus mushrooms is ambiguous, did the authors align the sequence of ITS in order to make sure you were working with pure Pleurotus ostreatus strains

What do you mean by “pleasant in structure”? please use scientific expressions.

Line 214 this section should be Results and Discussion?

L241: The variety between strains might be more directly perceived from a bar chart rather than a table like Table 2.

  • According to the results of the genetic diversity analysis, strains 45 and 89 were probably the same strain, and 95 and 96 might also be the same strain. However, the glucan content did not show the similarity of these strains. Why is it?
  • The statistical results should be re-organized in a more logical and coherent way. What is statistical model of the data analyses.
  • A ANOVA table is suggested to present the source of variation and the mean square, percentage of variance explained by the model (R2), coefficients of variation (CV), etc.
  • The variety between strains might be more directly perceived from a bar chart rather than a table like Table 2. Nevertheless, what criteria did you use to sort the data of Table 2, neither the no. of strains nor the content of glucans. Anyway, it is difficult for readers to catches the key data of your study.
  • Table 3 did not give extra information compared to Table 2, which could be combined the two tables, otherwise mentioned in texts.
  • Too many short paragraphs were listed, which should be re-organized and combined.
  • Results about the glucan content and the genetic diversity are the two independent parts of this study. Discussions should be mentioned essentially to explain why the authors conducted them together and logical association between the two parts.

Author Response

Dear reviewer,

Thank you very much for your comments, which after incorporation have definitely improved the quality of our manuscript. Below we answer to specific comments in points.

--- However the two parts of the results are lack of association

///Dear reviewer,

Thank you for reviewing our manuscript and for your comment. We tried to create a connection between the individual chapters, changing the order of chapters in the manuscript.

The problem arose as follows. The study was developed by two independent research teams. The first solved the primary task (completely rational), to identify the same strains of oyster mushroom and thus reduce the number of samples and thus the financial and time difficulty of the experiment. The biological material was from different sources and under the different designations, we did not know which samples were parallel. However, the genetics team had to optimize the entire methodology to meet this goal. After the successful implementation of the experiment, we considered it appropriate to publish it together, originally with the main part - the analysis of glucans in the fruiting bodies of oyster mushroom. The genetics team found almost complete authenticity of the oyster mushroom strains, as the parallel samples were evaluated only 4 of all the analyzed samples. Therefore, we decided to keep all the strains in order.

The second research team was focused on the main part of our study, and that the identification of glucans in 59 production strains, individually in the stipes and in the caps. To our knowledge, such an extensive study is unparalleled.

The absent connection between the two individual chapters in the manuscript argues that one team did not want to interfere in the work of the other team. However, we have made a correction.

Changes made: change of chapters' order; addition of explanatory and connecting phrases; renumbered tables, chapters, and literature; strengthening and complementing the genetic part

--- Line35: a number of

--- Line 37 which are nontoxic and biodegradable

--- Line60-62: “To this day, have been conducted…”, Please rephrase the sentence.

/// Thank you, your comments have been incorporated.

---Lines: What do you mean by “oyster mushroom production strains”, are they commercial cultivars or strains can produce mushrooms? If you meant commercial cultivars, I doubt you might mix the strains of different species of Pleurotus mushrooms. Base on the fact that the taxonomy of Pleurotus mushrooms is ambiguous, did the authors align the sequence of ITS in order to make sure you were working with pure Pleurotus ostreatus strains

/// As we are reporting in methodology, oyster mushroom strains of various sources were used in the experiment. Comercially available and commonly used strains have been used, as well as isolates of wild mushrooms. All were tested for production potential under controlled conditions. The ITS technique was used to determine the quality and quantity of isolated DNA, (it is also written in the methodology) not to distinguish fungal strains. The CDDP technique was used to distinguish fungal strains.

---What do you mean by “pleasant in structure”? please use scientific expressions.

---Line 214 this section should be Results and Discussion?

/// Thank you, your comments have been incorporated.

---L241: The variety between strains might be more directly perceived from a bar chart rather than a table like Table 2.

/// Dear reviewer, thank you for your comment, we agree with you, but no correction is possible to make. We tried to insert the data into a bar graph. In the case of a combination of three parameters (TG, AG, BG) in 59 production strains, the result is a large graph with a small visible size of the presented values, which is not possible to read in the printed version of the publication. The solution to this problem would not be even to turn the page horizontally. In the case of an online publication, the reader can zoom the graph, but at one point, it would not be possible to see all the values, so the meaning of making these changes is lost. In this case, the table seems to be the most appropriate way of presentation of the values. However, if you insist on a change, we can make it.

---According to the results of the genetic diversity analysis, strains 45 and 89 were probably the same strain, and 95 and 96 might also be the same strain. However, the glucan content did not show the similarity of these strains. Why is it?

---The statistical results should be re-organized in a more logical and coherent way. What is statistical model of the data analyses.

---A ANOVA table is suggested to present the source of variation and the mean square, percentage of variance explained by the model (R2), coefficients of variation (CV), etc.

/// Dear reviewer,

In the part dealing with glucan analysis (Table 2), a column was added in the original manuscript dividing all strains into homogeneous groups at Method: 95.0 percent LSD based on β-glucan content, which are considered the most important substance in Pleurotus ostreatus. Based on the great variability of the data, 9 homogeneous groups were created. It is possible, but not meaningful, to present these results. After careful consideration, together with the colleagues from the research team we decided before sending the manuscript that we would not use statistics when evaluating strains. If we want to present the difference in each other in production strains, for example in the content of β-glucans, it would be necessary to eliminate possible errors by certainty, for example by subjectively separating the stipe from the cap. We suppose that our results are correct and objective, but with such a low difference in the glucan content between strains, we do not consider it correct to present them using statistics, given the small errors in manual performance. In case of your interest, we can send you the values immediately.

Another case is the evaluation of the glucan content in the stipe vs. in the. Here the results are so different that even after taking into account any small errors of the manipulator, we consider them highly objective and therefore we present them in the work.

The following paragraph was added to the text of the manuscript: In the study, we did not perform a statistical analysis of glucan content between strains on purpose. The reason is that the result could be distorting and misleading. Data variability in individual glucan content is large and the differences between strains are low (Table 2). The analyzes were performed accurately and the results are objective. However, in the case of Pleurotus ostreatus, it is not possible to separate the fruiting body into a stipe and a cap as precisely as, for example, in the species Agaricus bisporus. We therefore consider it confusing to present the results through statistics.

--- The variety between strains might be more directly perceived from a bar chart rather than a table like Table 2. Nevertheless, what criteria did you use to sort the data of Table 2, neither the no. of strains nor the content of glucans. Anyway, it is difficult for readers to catches the key data of your study.

/// Thank you for the comment. We forgot to emphasize that the table is ranked according to the priority "β-glucan in the stipe", which is the most important factor from the point of view of the manufacturing industry. We added a note to the text.

Presenting the results through the graphs might be a more ideal way of the interpretation, but not at the 59 strains of Pleurotus ostreatus. We have explained the fact in more detail in the text above.

--- Table 3 did not give extra information compared to Table 2, which could be combined the two tables, otherwise mentioned in texts.

/// With your consent, we will keep the table in the manuscript. We consider it important, becauce it presents average values. Due to our experiences we can confirm that the reader is firtsly observing short paragraphs. Our study is unique because of wide range of monitored strains, which we are dividing into strains with a high and low beta-glucan content. To discuss our work with other authors, Table 3 is necessary (we do not want to combine it with Table 2). After an extensive study of new and older scientific sources, we can say that most authors report only the average values of analyzed species . From our point of view, these are two separate "categories" of results.  

--- Too many short paragraphs were listed, which should be re-organized and combined.

/// Thank you, your comments have been incorporated.

--- Results about the glucan content and the genetic diversity are the two independent parts of this study. Discussions should be mentioned essentially to explain why the authors conducted them together and logical association between the two parts.

/// Dear reviewer, thank you for your comments. We agree and the changes have been incorporated in the manuscript.

We firmly believe that we have fulfilled your expectations by editing the manuscript.

Sincerely, the authors of the work.

Reviewer 4 Report

This is an interesting work based on sound experimental approach. 

However, to my opinion, major revisions should be made:

(a) extensive editing of English language is indeed required throughout the whole manuscript,

(b) in the Introduction, lines 70-71, only the mechanism regarding the direct immunomodulating effect of beta-glucans is mentioned, there is also an indirect effect through their fermentation in the colon and SCFAs production and this should be mentioned as well, 

(c) the Results section should be renamed to Results & Discussion as a large part of it (i.e. lines 281-316) is dedicated to commenting on the results in relation to previous findings,

(d) are the CDDP results somehow correlated to glucan content and/or fertility potential of the strains examined? This is a critical as well as interesting question, there should be an attempt to cope with it in the manuscript.   

Minor revisions:

- the legends are usually placed below the Figures.

- the text between lines 172-177 is redundant, normally, only the high-tech equipment/infrastructure is included in the manuscript  

Author Response

Dear reviewer,

Thank you very much for your comments, which after incorporation have definitely improved the quality of our manuscript. Below we answer to specific comments in points.

--- extensive editing of English language is indeed required throughout the whole manuscript,

///  Thank you for your comment, the English language has been reviewed and edited by 2 certified language instructors. Some adjustments have been made.

--- in the Introduction, lines 70-71, only the mechanism regarding the direct immunomodulating effect of beta-glucans is mentioned, there is also an indirect effect through their fermentation in the colon and SCFAs production and this should be mentioned as well, 

/// Thank you for your comment, we have incorporated a brief mention about given thing into the theoretical part. Also, the new citation was incorporated.

--- the Results section should be renamed to Results & Discussion as a large part of it (i.e. lines 281-316) is dedicated to commenting on the results in relation to previous findings,

/// Thank you for your remark, we renamed the chapter.

- are the CDDP results somehow correlated to glucan content and/or fertility potential of the strains examined? This is a critical as well as interesting question, there should be an attempt to cope with it in the manuscript.   

///Dear reviewer,

Thank you for reviewing our manuscript and for your comment. We tried to create a connection between the individual chapters, changing the order of chapters in the manuscript.

The problem arose as follows. The study was developed by two independent research teams. The first solved the primary task (completely rational), to identify the same strains of oyster mushroom and thus reduce the number of samples and thus the financial and time difficulty of the experiment. The biological material was from different sources and under the different designations, we did not know which samples were parallel. However, the genetics team had to optimize the entire methodology to meet this goal. After the successful implementation of the experiment, we considered it appropriate to publish it together, originally with the main part - the analysis of glucans in the fruiting bodies of oyster mushroom. The genetics team found almost complete authenticity of the oyster mushroom strains, as the parallel samples were evaluated only 4 of all the analyzed samples. Therefore, we decided to keep all the strains in order.

The second research team was focused on the main part of our study, and that the identification of glucans in 59 production strains, individually in the stipes and in the caps. To our knowledge, such an extensive study is unparalleled.

The absent connection between the two individual chapters in the manuscript argues that one team did not want to interfere in the work of the other team. However, we have made a correction.

Changes made: change of chapters' order; addition of explanatory and connecting phrases; renumbered tables, chapters, and literature; strengthening and complementing the genetic part

--- the legends are usually placed below the Figures.

/// Thank you, your comments have been incorporated.

--- the text between lines 172-177 is redundant, normally, only the high-tech equipment/infrastructure is included in the manuscript  

/// Dear reviewer, based on the previous requests of MDPI reviewers at previously published articles, we have decided to keep the information in the manuscript. The previous reviewers required a detailed specification of the entire infrastructure which was used during the experiments.

We firmly believe that we have fulfilled your expectations by editing the manuscript.

Sincerely, the authors of the work.

Round 2

Reviewer 1 Report

Dear authors;

After a second revision or the article entitled "Analysis of biochemical and genetic variability of Pleurotus ostreatus based on the β-glucans and CDDP markers" under the journal number jof-1701829.

I still want further corrections to improve the quality of the article and will be accepted in a scientific journal with a high impact factor (JOF)
I see that the authors have improved the version and have answered and corrected most of the remarks except always the genetic part and the use of CDDP markers are not well illustrated nor the exact number of primers, there is not a table which illustrates the parameters of the molecular polymorphism, the authors can even improve the statistics in this part, they add an ACP pa example.
this part needs to be improved you can consult several articles on CDDPs to take Haffar et al., 2022 as an example; Aouadi et al., 2019
you can even add an electrophoresis example for CDDPs
Also there are missing threads for this part

Good luck

Author Response

////Dear Reviewer,

Thank you for your comments. Our colleagues from the genetics team have attempted to improve the manuscript again. We hope we reached your expectations.

 In addition to the changes in the "genetic part," we revised the genetic part and had the grammar re-checked by a certified interpreter.

Yours sincerely,

Authors

Reviewer 3 Report

Line 156-160 You did grow mushroom in four replicates and two culivation cycles. How did you sample the mushroom caps and stipes? Did you put the materials of the four bottles toghter or test them seperately? please clarify in M & M

Line 231-233 please rephrase, not understandable.

The authors were not sure if the the strains were designated with the same names. The question is still not answered that if these strains were all Pleurotus ostreatus rather than P. florida or P. cornucopiae.

Line 247-248, please rephrase the sentence, too many "and"s

Line252,what do you mean by Chapter 2?

Line 253, not understandable, what do you mean by original and not ajusted.

Did you observe average phenotypes of mushrooms of the parallel samples ? If yes, I would expect a similar content.

LIne 299-301 the publications about the variablility of oyster mushrooms are not scarce. 

Line 313-314 you could delete the sentence "The values are shown in the table below".

what do you mean by chapter 1

If the glucan content of oyster mushrooms was generally higher than that of yeast? What are the priorities of the glucans in oyster mushrooms compares to that of yeast? Please discuss about this point.

Again I strongly recommend the authors to improve the English writing by native English speakers.

Author Response

Dear reviewer,

Thank you very much for your comments, which after incorporation have definitely improved the quality of our manuscript. Below we answer to specific comments in points.

Line 156-160 You did grow mushroom in four replicates and two culivation cycles. How did you sample the mushroom caps and stipes? Did you put the materials of the four bottles toghter or test them seperately? please clarify in M & M

///// Dear Reviewer, thank you for your comment. Yes, it was analyzed an average sample of about 45 fruiting bodies divided into the stipe and cap. We present this in the chapter 2.5. Determination of glucans.

Line 231-233 please rephrase, not understandable.

///// Thank you, the comment has been incorporated to the manuscript.

The authors were not sure if the the strains were designated with the same names. The question is still not answered that if these strains were all Pleurotus ostreatus rather than P. florida or P. Cornucopiae.

///// Our explanation is probably confusing. In part Materials and Methods, in part 2.1. Biological material is listed the original designation from the source, the genetic bank. The organizations state that it was exclusively Pleurotus ostreatus. The mushrooms were grown in identical conditions at a temperature of 16 °C. At the same time, no morphological difference was observed between the fruiting bodies. We assume that all the strains were Pleurotus ostreatus.

Line 247-248, please rephrase the sentence, too many "and"s

/////Thank you, the comment has been incorporated into the manuscript.

Line252,what do you mean by Chapter 2?

///// That means, the original aim was not to analyze strains potentially identical according to CDDP. However, since it was only about 2 pairs, we left all 4 strains in the research. Subsequently, we observed in chapter 2. Concentration of glucans in production strains of oyster mushroom (Pleurotus ostreatus) whether the glucan content in these strains is similar or the same.

Line 253, not understandable, what do you mean by original and not ajusted.

/////This means that despite the fact that strains PO-95 and PO-96; and PO-45 and PO-89 have identical CDDP profiles, we did not make only two average samples from these samples but we kept all 4 samples in the study. The results were not adjusted by averaging.

Did you observe average phenotypes of mushrooms of the parallel samples ? If yes, I would expect a similar content.

/////Dear Reviewer, we have the same opinion as you. However, in the discussion, we are commenting that the different results are probably due to the fact that it is not always possible to separate the stipe from the cap in Pleurotus ostreatus. At the same time, the deviations of specific strains are not significantly different. For example, strain PO-89 contained 44 % BG in stipe and strain PO-45 42 % BG in stipe. For the pair PO-96 and PO-95, the values were 45 % and 41 %. We suppose that these are related strains.

LIne 299-301 the publications about the variablility of oyster mushrooms are not scarce. 

///// Thank you, the comment has been incorporated to the manuscript.

Line 313-314 you could delete the sentence "The values are shown in the table below".

/////With your permission, we will keep the phrase in the manuscript. From our previous experience, MDPI usually requires a reference to a table in the text.

what do you mean by chapter 1

///// It is a connection of the text to the CDDP fingerprinting chapter of Pleurotus ostreatus.

If the glucan content of oyster mushrooms was generally higher than that of yeast? What are the priorities of the glucans in oyster mushrooms compares to that of yeast? Please discuss about this

/////Dear Reviewer, we must admit that this comment caused us the most difficulties. We are not sure if we understand your question correctly. Are you asking about the role of β-glucans in yeast versus oyster mushrooms, please? Or what kinds of benefits do β-glucans isolated from yeast versus from oyster mushroom have to humans, please?

In yeast:

The yeast cell wall consists of three main groups of polysaccharides: mannose polymers covalently bound to peptides (mannoproteins, representing about 40 % of the cell wall dry matter), glucose polymers (β-D-glucans, namely β-1,3-D-glucans and β-1,6 -D-glucans forming about 60 % of the dry matter of the wall mass) and polymers of N-acetylglucosamine (chitin, about 2 % of the dry matter of the cell wall).

Yeast β-D-glucans can be divided into linear and branched. Linearly, containing about 1,500 glucoses linked by β-1,3 glycosidic bonds, they are the most common glucans with only one type of glycosidic bond, which are found in many types of fungi, where they form the basic structure of the cell wall. Of these, β 1,3 glucans of the yeast Saccharomyces cerevisiae are the most studied. β 1,3 glucans represent 65-90 % of the total glucans in the cell walls of fungi.

β-1,6 glucans have also been identified in these yeasts, which are most abundant in lichens. They usually contain a short chain consisting of about 150 glucoses. They form about 15 % of β-D-glucans. As the components of the cell wall, they are important in flocculation so-called killer toxin receptor, and systems that respond to the changes in the environment, the medium in which the yeast grows and allow them to adapt, detoxify harmful substances.

Branched β-glucans containing β 1,3 glycosidic bonds and branching β 1,6 have also been demonstrated in yeast.

The structure studies of fungal glucans were performed with polysaccharides produced by S. cerevisiae during protoplast regeneration. Yeast glucans had a microfibril structure approximately 4 μm in length and 20 nm in diameter (Kreger and Kopecká, 1975; Kopecká and Kreger, 1986). This microfibrillar structure has also been observed in β 1,3 glucans synthesized in vitro by yeast extracts (Larriba et al., 1981). While the formation of linear β 1,3 glucans is relatively rare, β 1,3 glucans synthesized in vitro by S. cerevisiae membrane fractions have been branched.

Considering the microfibrillar structure, β 1,3 glucans have an important role in the structure, resp. cell wall architecture and resilience. Inhibition of β 1,3 glucan synthesis leads to either cell lysis or to the changes in cell morphology. S. cerevisiae protoplasts can be prepared by the action of β 1,3 glucanase.

How can the strength and resistance of the fungal cell wall be explained? Recent results with S. cerevisiae suggest that the strength and resistance of the yeast cell wall is explained by the presence of bonds between chitin and β 1, - glucans, as well as between glycoproteins, β 1,6-glucans and β 1,3-glucans.

Table: β-glucan content from various sources. β-glucan usually occurs in cereals in low concentrations (~ 5 % w/v). The amounts of these polymers vary considerably in microorganisms. Optimized yields of β-glucan extracted from baker's yeast (Saccharomyces cerevisiae) provided only 5 % -7 %, but Euglena can accumulate β-glucan intracellularly to more than 90 % [23]. The β-glucan content of seaweed depends mainly on the species. Durvillaea antarctica contains 33 % and < 5 % β-glucan [24]. The β-glucan content of fungi also varies widely, from about 3.1 % to 46.5 % [25].

In oyster mushrooms:

A polysaccharide containing a branch consisting of only a single β 1,3-linked glucosyl group was found in Pleurotus florida and Lentinula edodes. In Lentinula (Lentinus) squarrosulus, a polysaccharide that contains a glucan chain with both β 1.3 and β 1,6 bonds and a β 1,6 branch is formed by a single molecule of glucose.

Pleuran is part of the cell walls.

β-glucans are part of the cell walls of basidiomycetes.

β-glucans are referred to as biological response modifiers. These are natural substances - polysaccharides, which appear in cell walls, seaweed, fungi (shiitake, reishi, coriolus) or, for example, yeast. The most common preparations are those containing β-glucan isolated from the bakery and beer yeast Sachcaromces cerevisia. These are the most natural for our body and also the most biologically active are the β-glucans with the basic chain of glucose molecules connected by a 1,3 bond and with the glucose side chain in position 1,6.

////Dear reviewer, thank you for the recommendation to the language corrections. Another licensed interpreter double-checked the English language, and his changes were incorporated into the manuscript. The translator ensured that the manuscript is grammatically correct. We are hopeful that everything is fine now.

We firmly believe that we have fulfilled your expectations by editing the manuscript.

Sincerely, the authors of the work.

Reviewer 4 Report

Please see comments in the attached file 

Author Response

Dear reviewer,

Thank you very much for your comments, which after incorporation have definitely improved the quality of our manuscript. Below we answer to specific comments in points.

Abstract

Lines 24-27: the statement is not clear, the reader should go down to the results & discussion section in order to understand the meaning to a certain degree. An abstract should clear and autonomous, therefore, you are kindly requested to rephrase for reasons of clarity.
/////Dear Reviewer, thank you for your comment. We have tried to edit the abstract according to your comment. We highlighted the fact that the stipes contained more strains than caps, the strains with the lowest as well as the highest BG content in the stipes and strains with the same CDDP profile. The maximum range of the abstract is already exceeded. We would like to mention that, in the last article, the reviewers recommended that in the abstract we should not state more facts than is necessary.  The reader should be forced to read the abstract.

Introduction

Lines 37-38: the statement is not correct, not all heterogeneous fungal polysaccharides are β-glucans, so please correct.

Lines 48-49: “The major role of biological activities in fungi is performed by β-glucans”, this is not correct. β-glucans are important bioactive polysaccharides but there are several fungal bioactive compounds (i.e.polyphenols) contributing to the manifestation of the health-promoting properties of fungi. So, please correct accordingly.

/////Thank you for your remark. The sentence is based on a scientific source and its intention was not to say that beta-glucans are the only compounds with health-promoting properties, but are the main ones.  Your comment has been incorporated into the manuscript.

Line 57: “It results in the digestion of fungal cell wall” instead of “It results digestion of fungal cell wall”.

Lines 68-69: please change “civilization related disease” to “lifestyle related diseases”.

Lines 74-84: this part needs extensive language editing

/////Thank you, the comment has been incorporated into the manuscript.

Line 101: “…in the glucans content of in the stipes and the caps” should change in “…in the glucans content of the stipes and caps”

Materials & Methods

Line 119: please change “CDDP analyse” into “CDDP analysis”.

Results & Discussion

Lines 234-236: please restructure the phrase as follows “As already mentioned, there is a variability between individual oyster mushroom production strains when different aspects of production potential as well as genetic polymorphism are compared”.

Line 244: please change “UPGMA analyze” into “UPGMA analysis”.

Lines 293: Pleurotus ostreatus belongs to the important fungi instead of “fungus”.

Lines 332-333: please rephrase, this structure does not make any sense.

Lines 340: please change “From our observations results…” into “According to our observations, it is clear…”

Lines 411: “analyses” instead of “analyzes”.

Conclusions

Lines 461: please change “…, it is possible to directly increase the nutritional value…” to “…, is possible to directly increase the nutritional value…”.

Lines 463: please change “we recommend in further studies…” to “we recommend further studies…”.

/////Thank you, the comment has been incorporated into the manuscript.

Concluding, I strongly recommend an additional round of English language editing throughout the whole manuscript before resubmission.

///Dear Reviewer, thank you for the recommendation to the language corrections. Another licensed interpreter double-checked the English language, and his changes were incorporated into the manuscript. The translator ensured that the manuscript is grammatically correct. We are hopeful that everything is fine now.

We firmly believe that we have fulfilled your expectations by editing the manuscript.

Sincerely, the authors of the work.

This manuscript is a resubmission of an earlier submission. The following is a list of the peer review reports and author responses from that submission.